# Endoscopic Submucosal Dissection in Patients with Early Gastric Cancer in the Remnant Stomach

**DOI:** 10.3390/diagnostics12102480

**Published:** 2022-10-13

**Authors:** Mai Murakami, Takuto Hikichi, Jun Nakamura, Minami Hashimoto, Tsunetaka Kato, Ryoichiro Kobashi, Takumi Yanagita, Rei Suzuki, Mitsuru Sugimoto, Yuki Sato, Hiroki Irie, Mika Takasumi, Tadayuki Takagi, Yuko Hashimoto, Masao Kobayakawa, Hiromasa Ohira

**Affiliations:** 1Department of Gastroenterology, Fukushima Medical University School of Medicine, Fukushima 960-1295, Japan; 2Department of Endoscopy, Fukushima Medical University Hospital, Fukushima 960-1295, Japan; 3Department of Diagnostic Pathology, Fukushima Medical University School of Medicine, Fukushima 960-1295, Japan; 4Medical Research Center, Fukushima Medical University, Fukushima 960-1295, Japan

**Keywords:** remnant stomach, endoscopic resection, endoscopic submucosal dissection, gastrectomy, gastric cancer

## Abstract

Endoscopic submucosal dissection (ESD) in patients with early gastric cancers (EGCs) in the remnant stomach is technically difficult, owing to the limited space and fibrosis under the suture lines and anastomoses. Conversely, ESD for patients with EGCs in the remnant stomach is less invasive and provides better quality of life than completion total gastrectomy. To clarify the effectiveness and safety of ESD, we reviewed the medical records of patients with EGCs in the remnant stomach who underwent ESD between July 2006 and October 2020 at our institution. All identified patients were included in the analysis. Of 25 patients with 27 lesions, the *en bloc* and R0 resection rates were 88.9% and 85.2%, respectively. Neither perforation nor postoperative bleeding was observed. During a median follow-up period of 48 (range, 5–162) months, the 5-year overall survival rate was 71.0%, whereas the 5-year cause-specific survival rate was 100%. No obvious differences in the outcomes of procedures with suture line involvement and without suture line or anastomosis involvement were noted. In conclusion, ESD was effective and safe in patients with EGCs in the remnant stomach despite the suture line involvement.

## 1. Introduction

Remnant gastric cancer (RGC) is defined as cancer arising in the remnant stomach after previous gastrectomy [1]. The cumulative incidents of RGC in patients who underwent pyloric gastrectomy with Billroth-I reconstruction for gastric cancer were reported to be 3.7% and 5.4% at 10 and 20 years, respectively [2]. In the past, RGCs were often detected in advanced cancer; however, in recent years, with the spread of endoscopic surveillance after gastric surgery, the number of RGCs detected in early-stage cancer has been increasing [3].

Completion total gastrectomy is typically performed for patients with RGC. However, completion total gastrectomy poses risks of surgical complications in 46.6% of patients, including intra-abdominal abscess, ascites, and wound infections, and possibly deteriorates quality of life due to the absence of the function of the stomach, with a reported 3-year postoperative survival rate of 63.4% [4]. Conversely, the efficacy of endoscopic submucosal dissection (ESD) has been reported for the treatment of intramucosal RGC, which does not require lymph node dissection and allows the preservation of the stomach [5,6,7,8,9,10]. However, ESD for patients with RGC is problematic owing to the difficulty of endoscopic manipulation in the limited space and the difficulty of dissection due to submucosal fibrosis at suture lines and anastomoses [11,12]. Although there are several reports on the efficacy and safety of ESD for patients with RGC [5,6,7,8,9,10], the long-term prognosis was not sufficiently evaluated. Therefore, our study aimed to clarify the precaudal efficacy and safety of ESD for patients with RGCs according to clinicopathological characteristics and long-term prognosis.

## 2. Materials and Methods

### 2.1. Patients

We conducted a single-center, retrospective cohort study. Patients who underwent ESD for the first time for early-stage RGCs at Fukushima Medical University Hospital between July 2006 and October 2020 were included in the study. Patients with second and subsequent ESDs for early-stage RGCs were excluded.

Patient information data were retrospectively collected from electronic medical records and endoscopic databases as of October 2020. If follow-up was conducted at another institution, the responsible physician contacted the patients to determine if they were alive, and in case of death, the cause of death was ascertained from the family.

### 2.2. Practice of ESD and Follow-Up

The following are the indications for ESD in patients with RGC at our institution, which are the same as those in conventional early gastric cancer (EGC): an intramucosal differentiated-type adenocarcinoma without ulcerative findings, an intramucosal differentiated-type adenocarcinoma of 3 cm or less with ulceration, and an intramucosal undifferentiated-type adenocarcinoma of 2 cm or less without ulceration [13].

All ESDs were performed using video endoscopes (GIF-Q260J, GIF-2TQ260M, GIF-H290TI; Olympus Medical Systems Co., Ltd., Tokyo, Japan) and a standard video endoscope system (EVIS LUCERA or EVIS LUCERA ELITE; Olympus), the same as previous reports [14,15,16,17,18]. First, we marked circumferentially 5 mm outside the lesion using a Dual Knife (KD-650L; Olympus). Next, a mixture of 0.4% sodium hyaluronate (MucoUp; Boston Scientific Japan, Tokyo, Japan) and glycerol (Chugai Pharmaceutical Co., Ltd., Tokyo, Japan) was injected into the submucosa using an injection needle. Then, mucosal incision and submucosal dissection were performed using a Dual Knife, an IT Knife 2 (Olympus), or an SB Knife Jr. (SB Kawasumi, Tokyo, Japan); if needed, injectant was added into the submucosa. A hemostatic forceps device (Coagrasper; Olympus) was used for intraoperative prophylactic coagulation and hemostasis. A VIO300D or VIO3 (ERBE Elektromedizin GmbH, Tübingen, Germany) was used as a high-frequency generator. All ESDs were performed under sedation with midazolam with pentazocine or with propofol with pentazocine. All ESDs were performed by a board-certified fellow of the Japan Gastroenterological Endoscopy Society. However, until 2010, some cases were performed by non-experts, with <100 cases of ESD under the supervision of an expert. Immediately after the ESD procedure, computed tomography (CT) was performed only when endoscopically suspected perforation was detected during ESD.

After ESD, annual endoscopic follow-up was performed in cases of curative resection. Additional surgery was recommended for patients who underwent non-curative resection. However, CT and endoscopy were performed once or twice a year if the patient refused surgery.

### 2.3. Outcomes

ESD outcomes, including procedure time, *en bloc* resection and R0 resection rates, endoscopic curability, and procedure-related adverse events, were evaluated. Moreover, the local recurrence rate, distant metastasis rate, metachronous cancer incidence rate, 5-year overall survival (OS), and 5-year cause-specific survival were evaluated as long-term prognostic measures.

Pathological staging of resected specimens sliced at 2 mm intervals was determined according to the Japanese Classification of Gastric Carcinomas [1]. Tumor diameter and depth; histological and microscopic types; ulcer findings; lymphovascular invasion (LVI), such as lymphatic or venous invasion; and horizontal/vertical resection margins were microscopically determined. Initially, LVI was determined using hematoxylin-and-eosin-stained sections, and if LVI was suspected, D2-40 and CD34 staining was additionally performed [19]. Procedure time was defined as the time from the start of mucosal incision to the end of submucosal dissection. R0 resection was defined as *en bloc* resection and the absence of cancer at the horizontal margins (HMs) and vertical margins (VMs) [9]. Curative resection was defined as an R0 resection with no LVI and (1) an intramucosal differentiated-type adenocarcinoma without ulcerative findings, (2) an intramucosal differentiated-type adenocarcinoma of ≤3 cm with ulcerative findings, (3) an intramucosal undifferentiated-type adenocarcinoma of ≤2 cm without ulcerative findings, and (4) a superficial submucosal (<500 µm from the muscularis mucosae) differentiated adenocarcinoma. All other cases were defined as non-curative resection.

Regarding procedure-related adverse events, perforation was defined as endoscopic confirmation of the abdominal cavity or free air on CT [15]. Postoperative bleeding was defined as the presence of hematemesis or black stools after ESD and active bleeding or exposed blood vessels on endoscopy [15].

Serum anti-*Helicobacter pylori* (*H. pylori*) immunoglobulin G (IgG) antibody (Otsuka Pharmaceutical Co., Ltd., Tokyo, Japan) was used to diagnose *H. pylori* infection. The results were considered positive when the concentration of serum anti-*H. pylori* IgG antibody was 10 U/mL or higher [17]. The Charlson comorbidity index (CCI) was used to evaluate risk due to comorbidities, and scores of 0, 1–2, 3–4, and 5 or more were defined as low, medium, high, and very high risk, respectively [20].

### 2.4. Statistical Analysis

Continuous values of patients’ characteristics were presented as medians with ranges. Survival rates were estimated using the Kaplan–Meier method. Clinical outcomes were analyzed using Fisher’s exact test for categorical variables and *t*-test or the Mann–Whitney *U* test for continuous variables. *P* < 0.05 was considered statistically significant. The analyses were performed using SPSS software (version 28 for Windows; IBM Corp, Armonk, NY, USA).

## 3. Results

### 3.1. Patient and Lesion Characteristics

Twenty-seven RGC lesions in 25 cases were identified, and all of them were included in the analysis (Table 1). The median age was 74 years, and 14, 9, and 2 patients had low, medium, and high CCIs, respectively. Twenty-one patients (84%) had a previous distal gastrectomy, and 3 (12%) had a proximal gastrectomy. Sixteen, three, and two patients had Billroth-I, Billroth-II, and Roux-en-Y anastomoses for the reconstruction of gastrectomy. The causes of gastrectomy were gastric cancer, benign diseases of the stomach or duodenum, and intraductal papillary mucinous carcinoma in 20 (80%), 4 (16%), and 1 patient (4%), respectively. The median time from previous gastrectomy to index ESD was 168 (range, 24–552) months. The time from previous gastrectomy to index ESD was significantly longer in patients with a history of benign gastroduodenal disease (median, 390 (range, 264–552] months) than in those with a history of gastric cancer (median, 156 (range, 24–384) months) (*p* = 0.003).

The characteristics of the 27 RGC lesions are summarized in Table 2. Seven (25.9%), three (11.1%), five (18.5%), and twelve (44.4%) lesions were located in the anterior wall, greater curvature, posterior wall, and lesser curvature, respectively. Twelve lesions (44.4%) were involved in the gastrectomy suture line (Figure 1), and one lesion (3.7%) was involved in the anastomosis from the previous gastrectomy.

### 3.2. ESD Outcomes

In two cases, the two lesions were resected *en bloc* as a single specimen. The median procedure time was 80 min; the *en bloc* resection rate was 88.9%, and the R0 resection rate was 85.2% (Table 3). Neither perforation nor postoperative bleeding occurred. The median tumor diameter of resected specimens was 12 mm; the median specimen diameter was 37.5 mm, and all lesions were differentiated adenocarcinomas. The curative resection rate was 81.5% (22/27), and five patients underwent non-curative resection. The comparison of the treatment outcomes of ESD according to the involvement in suture line or anastomosis is shown in Table 4. No statistically significant differences were noted in tumor and specimen diameters, procedure time, *en bloc* and R0 resection rates, or the curative resection rate between lesions on the suture line (*n* = 12) and lesions either on the anastomosis or on the suture line (*n* = 14). The reasons for the five non-curative resections were piecemeal resection in two cases, piecemeal resection and VMX in one case, positive HM in one case, and deep submucosa of tumor depth (SM2) and positive venous invasion in one case (Table 5). However, all five patients did not wish to undergo additional surgery and were followed up.

Although this study did not examine the ESD technique difficulty based on lesion circumference, the retroflex scope manipulation was required for lesions located in the lesser curvature, and the remnant stomach had a narrow working space, making it difficult to maneuver the scope in some patients. Additionally, lesions located in the lesser curvature were affected by staples and fibrosis because they were sometimes in contact with the suture line. Conversely, lesions located in the greater curvature had difficulty maintaining an efficient surgical field with blood when bleeding occurred intraoperatively.

The median diameter of the resected specimens and procedure time were 34 (range, 18–55) mm and 78 (range, 42–210) min, respectively, in 21 RGC lesions after distal gastrectomy. Conversely, RGCs after proximal gastrectomy were five lesions in three patients, and two of them had two lesions resected *en bloc* as a single section. The median diameter of the resected specimens in three patients with RGC after proximal gastrectomy was 44 mm (range, 40–82), and the median procedure time was 78 min (range, 42–210).

### 3.3. Long-Term Outcomes

The clinical course flow diagram of 25 patients is presented in Figure 2. The median follow-up period after ESD was 48 (range, 5–162) months (Table 6). During the follow-up period, an 83-year-old male patient who underwent non-curative resection due to a positive HM developed local recurrence 8 months after the initial ESD (Figure 3, Table 5). The patient underwent a second ESD for the recurrent lesion; however, he died of lung cancer 12 months after the second ESD.

The 5-year OS and cause-specific survival rates were 71.0% and 100%, respectively (Figure 4). During the observation period, 10 patients (40%) died of other causes: two of pneumonia, one of prostate cancer, one of hypopharyngeal cancer, one of lung cancer, one of drowning from a fall into the river, and four of unknown causes.

## 4. Discussion

In this study, ESD in patients with early stage RGC was safely performed without serious adverse events; however, the *en bloc* resection rate was 88.9%. Additionally, the 5-year cause-specific survival rate was 100%. To the best of our knowledge, there are only two papers evaluating 5-year OS rate and cause-specific survival rate in ESD for patients with RGCs.

ESD for patients with RGC is considered technically challenging, owing to the limited space in the stomach for endoscopic treatment and the difficulty of dissection when fibrosis is present on the suture line or at the anastomosis [11,12]. However, previous reports of ESD for patients with RGC have reported excellent results, with *en bloc* and R0 resection rates of 94.2–100% and 74.2–92.3%, respectively, since they were reported from ESD high-volume centers [5,6,7,8,9,10] (Table 7). Here, the *en bloc* and R0 resection rates were 88.9% and 85.2%, respectively, showing relatively low *en bloc* resection rates compared with those of previous reports. However, the three cases who underwent piecemeal resection were early-period cases after the introduction of ESD at our institution, which may have been due to the lack of skill of the surgeons.

In a comparison of cases involving the suture line versus those involving neither the suture line nor anastomosis, procedure time, *en bloc* and R0 resection rates, and curative resection rate in cases involving the suture line did not differ from those in cases involving neither the suture line nor anastomosis. Yabuuchi et al. [10] stated that the ESD of gastric cancer involving the suture line had similar *en bloc* resection and intraoperative perforation rates as the ESD of gastric cancer involving neither the suture line nor anastomosis. This may be because an experienced surgeon had carefully performed the ESD for patients with RGC on the suture. Conversely, Yabuuchi et al. [10] also stated that the *en bloc* resection rate was low, that the procedure time was long, and that the perforation rate was high in the ESD of gastric cancer involving anastomosis. However, we were unable to verify these findings because we had only one case of anastomotic lesion in this study.

Of the 25 cases, one case with positive HM developed a local recurrence cancer. This case was considered to be challenging to diagnose, owing to the location of the lesion in the cardia in addition to the remnant stomach, making a precise border diagnosis difficult to determine. This local recurrence cancer was also treated with ESD, and it was resected *en bloc* with curative resection. The other four patients, who underwent non-curative resection, chose follow-up without additional surgery; however, no death from gastric cancer was observed. Hatta et al. [21] reported the eCura system to evaluate the risk of lymph node metastasis using lymphatic invasion, a tumor diameter of >30 mm, tumor depth SM2, VM positivity, and venous invasion in cases of non-curative resection of ESD for gastric cancer. The patients were classified into low risk (lymph node metastasis rate: 2.5%), intermediate risk (6.7%), and high risk (22.7%) according to the score. Of the five patients in our study who underwent non-curative resection, four were classified into the low-risk group of the eCura system, and only one (a 79-year-old female) was classified into the intermediate-risk group. These may explain why there were no gastric cancer deaths even after non-curative resection, and the 5-year cause-specific survival rate was 100%. In contrast, the 5-year OS rate of the patients in the current study was 71.0%, which tended to be lower than that of previous reports. Iwai et al. [22] showed that CCI can be an important prognostic factor for OS in patients with EGC after ESD, and the OS rates in patients with low CCI scores (≤2) were significantly higher than those in patients with high CCI scores (≥3), regardless of age. In this study, however, only two cases had a CCI score of ≥3, and cases who died by drowning from a fall into the river and died from unknown causes were included; therefore, the association between OS and CCI was not evaluated. In this study, no gastric cancer deaths and only one case of local recurrence among the cases of non-curative resection were reported, which was promising in that it was possible to cure the patient with another ESD.

This study had several limitations. First, it was a single-center, retrospective study with a small number of patients. In particular, there was only one case of anastomotic lesion. Second, the anatomy of the suture line after gastrectomy differs between Lambert suture and mechanical double stapling, which may result in different layers of submucosal dissection. However, in this study, we could not confirm the suture method used in previous gastrectomies.

## 5. Conclusions

Herein, the *en bloc* resection rate was lower than that reported previously; however, the R0 resection rate was similar to that reported previously. Despite a lower number of patients, no procedure-related adverse events occurred, the 5-year cause-specific survival rate was 100%, and no death from gastric cancer occurred, which are novel characteristics of this study. Therefore, ESD for patients with early-stage RGC is a safe and effective treatment to avoid gastric-cancer-related deaths. Future studies are needed to evaluate ESD for early-stage RGC in a larger number of patients at multiple centers and with a longer follow-up period.

## Figures and Tables

**Figure 1 diagnostics-12-02480-f001:**
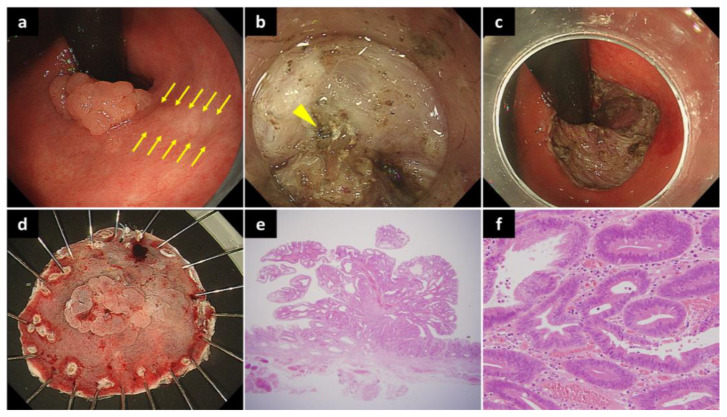
A lesion on the suture line: (**a**) An elevated gastric cancer has been observed on the suture line (between yellow arrows) in the lesser curvature of the remnant stomach. (**b**) During ESD, staples (yellow arrowhead) and massive fibrosis are present at the suture line. (**c**) The lesion is completely removed by ESD without perforation. (**d**) The resected specimen immediately after ESD. (**e**) Histological finding of the resected specimen reveals an intramucosal tubular adenocarcinoma with negative margins (HE staining). (**f**) Clusters of cancer cells forming glandular ducts were observed (HE staining). ESD, endoscopic submucosal dissection; HE, hematoxylin and eosin.

**Figure 2 diagnostics-12-02480-f002:**
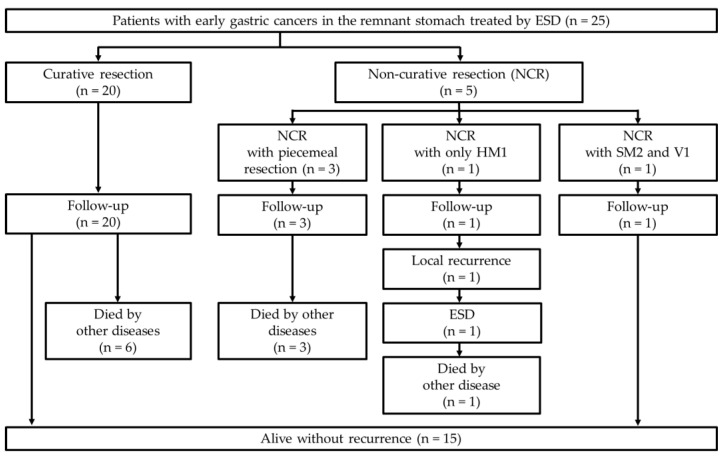
Clinical course flow diagram of the 25 patients. HM, horizontal margin; SM, submucosa; V, venous invasion; ESD, endoscopic submucosal dissection.

**Figure 3 diagnostics-12-02480-f003:**
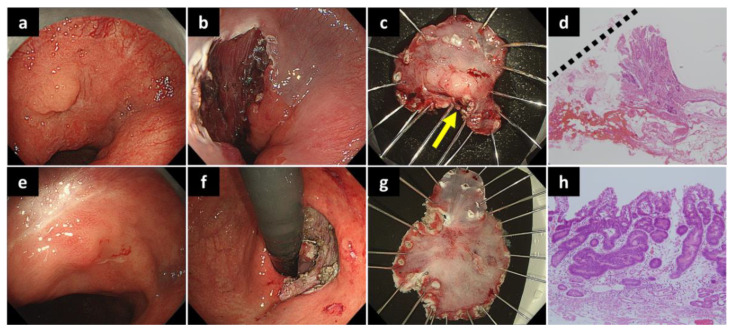
A case of local recurrence after ESD. (**a**) A 20 mm-sized elevated gastric cancer was observed by the cardia in the remnant stomach. (**b**) The lesion has been removed by ESD. (**c**) The margins of the lesion are close to the horizontal section. (**d**) Tumor cells are exposed on the resection surface (dashed line) (HE staining). The horizontal margin is evaluated as positive. (**e**) Eight months after ESD, a gastric carcinoma appears on the anorectal side of the post-ESD scar. It is judged to be a residual recurrence. (**f**) The local recurrence cancer is completely removed by the second ESD. (**g**) The resected specimen immediately after the second ESD. (**h**) The resected specimen shows a well-differentiated tubular adenocarcinoma. Horizontal and vertical margins are negative (HE staining). ESD, endoscopic submucosal dissection; HE, hematoxylin and eosin.

**Figure 4 diagnostics-12-02480-f004:**
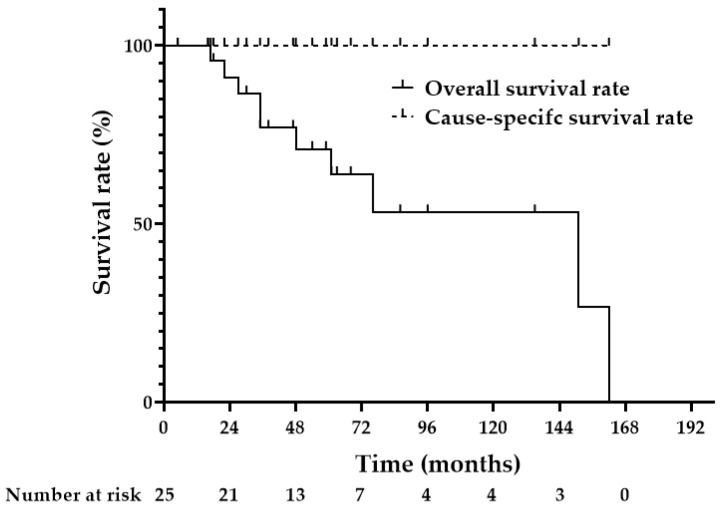
Overall and cause-specific survival rates. The 5-year overall and cause-specific survival rates are 71.0% and 100%, respectively.

**Table 1 diagnostics-12-02480-t001:** Patient characteristics (*n* = 25).

Patients/lesions, *n*	25/27 *
Age, median (range)	74 (63–89)
Sex, male/female	19/6
Height, median (range), m	1.59 (1.35–1.76)
Weight, median (range), kg	49.7 (35.2–74.6)
Body mass index, median (range)	20.0 (16.1–25.8)
Anti-*H. pylori* IgG antibodies (positive/negative/not tested)	8/16/1
Intake of antithrombotics, *n* (%)	6 (24)
Charlson comorbidity index (low/medium/high/very high)	14/9/2/0
Cause leading to gastrectomy, *n* (%)	
Gastric cancer	20 (80)
Gastric ulcer	2 (8)
Duodenal ulcer	1 (4)
Gastric polyp	1 (4)
IPMC	1 (4)
Type of gastrectomy, *n* (%)	
Distal	21 (84)
Proximal	3 (12)
Pancreaticoduodenectomy	1 (4)
Period from previous gastrectomy to index ESD, median (range), month	168 (24–552)

* Two patients each had two lesions simultaneously, which were resected as a single specimen. Anti-*H. pylori* IgG antibodies, Anti-*Helicobacter pylori* immunoglobulin G antibodies; IPMC, intraductal papillary mucinous carcinoma; ESD, endoscopic submucosal dissection

**Table 2 diagnostics-12-02480-t002:** Lesion characteristics (*n* = 27).

Lesion circumference, *n* (%)	
Anterior wall	7 (25.9)
Greater curvature	3 (11.1)
Posterior wall	5 (18.5)
Lesser curvature	12 (44.4)
Suture line involvement, *n* (%)	12 (44.4)
Anastomosis involvement, *n* (%)	1 (3.7)
Microscopic type, *n* (%)	
0–I	3 (11.1)
0–IIa	13 (48.1)
0–IIc	9 (33.3)
Mixed	2 (7.4)

**Table 3 diagnostics-12-02480-t003:** Treatment results of endoscopic submucosal dissection (*n* = 27).

Procedure time, median (range), min	80 (42–307)
*En bloc* resection, *n* (%)	24 (88.9)
R0 resection, *n* (%)	23 (85.2)
Tumor diameter, median (range), mm	12 (4-40)
Specimen diameter, median (range), mm	37.5 (18–82)
Histologic type, *n* (%)	
pap	2 (7.4)
tub1	23 (85.2)
tub2	2 (7.4)
Tumor depth, *n* (%)	
M	24 (88.9)
SM1	2 (7.4)
SM2	1 (3.7)
Ulcer finding, *n* (%)	2 (7.4)
Lymphatic invasion, n	0
Vascular invasion, *n* (%)	1 (3.7)
Horizontal margin, *n* (%)	
Positive or unevaluated	2 (7.4)
Negative	25 (92.6)
Vertical margin, *n* (%)	
Positive or unevaluated	2 (7.4)
Negative	25 (92.6)
Curative resection, *n* (%)	22 (81.5)
Procedure-related adverse events, *n*	0

pap, papillary adenocarcinoma; tub1, well differentiated tubular adenocarcinoma; tub2, moderately differentiated tubular adenocarcinoma; M, mucosa; SM1, superficial submucosa (tumor invasion is <500 μm from the muscularis mucosae); SM2, deep submucosa (tumor invasion is 500 μm or deeper from the muscularis mucosae).

**Table 4 diagnostics-12-02480-t004:** Comparison of the treatment outcomes of endoscopic submucosal dissection according to suture line or anastomosis involvement.

	Suture Line Involvement * (*n* = 12)	Involving Neither Suture Line Nor Anastomosis * (*n* = 14)	Anastomosis Involvement (*n* = 1)	*p* Value Between Lesions Involving Suture Line and Lesions Neither Involving Suture Line nor Anastomoses
Tumor diameter, median (range), mm	15.5 (7–40)	11.0 (4–25)	15	0.167
Specimen diameter *, median (range), mm	35.5 (22–55)	35.0 (18–49)	40	0.591
Procedure time *, median (range), min	93.5 (56–307)	69 (42–210)	80	0.381
*En bloc* resection, *n* (%)	11 (91.7)	12 (85.7)	1	1.000
R0 resection, *n* (%)	11 (91.7)	11 (78.6)	1	0.598
Curative resection, *n* (%)	11 (91.7)	10 (71.4)	1	0.330

* One case each of lesions involving the suture line and lesions involving neither suture line nor anastomosis had two lesions resected in a single section. These two cases and four lesions were excluded from the specimen diameter and procedure time.

**Table 5 diagnostics-12-02480-t005:** Characteristics of non-curative resection cases.

Age	Sex	Microscopic Type	Anastomosis or Suture Line Involvement	Depth	Tumor Diameter (mm)	Histology	Reasons of Non- Curative Resection	Additional Treatment	Recurrence	Prognosis
70	M	Mixed	None	M	-	tub1	Piecemeal resection	None	None	Died 162 months after ESD, due to prostate cancer
79	M	0–IIa	None	SM1	13	tub2	Piecemeal resection, VMX	None	None	Died 152 months after ESD, due to unknown cause
66	M	0–IIc	Suture line	M	28	tub1	Piecemeal resection	None	None	Died 48 months after ESD, due to unknown cause
79	F	0–IIa	None	SM2	25	tub2 > por	SM2, V1	None	None	Alive for 96 months
83	M	0–IIa	None	M	18	tub1	HM1	None	Local recurrence after 8 months (resected by ESD)	Died 12 months after the second ESD, due to lung cancer

M, male; F, female; M, mucosa; SM1, superficial submucosa (tumor invasion is <500 μm from the muscularis mucosae); SM2, deep submucosa (tumor invasion is 500 μm or deeper from the muscularis mucosae), tub1, well-differentiated tubular adenocarcinoma; tub2, moderately differentiated tubular adenocarcinoma; por, poorly differentiated adenocarcinoma; VMX, inconclusive vertical margin; V1, positive venous invasion; HM1, positive horizontal margin; ESD, endoscopic submucosal dissection.

**Table 6 diagnostics-12-02480-t006:** Long-term outcomes after endoscopic submucosal dissection.

Follow-up period, mean (range), month	48 (5–162)
Local recurrence, *n* (%)	1 (4)
Distant metastasis, *n*	0
Metachronous cancer, *n*	0
Death by gastric cancer, *n*	0
Death by other disease, *n* (%)	10 (40)

**Table 7 diagnostics-12-02480-t007:** Summary of previous reports.

First Author (Published Year)	No. of Patients	No. of Lesions	Mean or Median Age, Year	Lesion Located on the Suture Line, % (*n*)	*En bloc* Resection Rate, % (*n*)	R0 Resection Rate, % (*n*)	Adverse Events, % (*n*)	5-Year OS Rate, %	5-Year Cause-Specific Survival Rate, %
Hirasaki (2008) [5]	17	17	73.1	N/A	100 (17)	82.4 (14)	17.6 (3)	N/A	N/A
Takenaka (2008) [6]	30	31	73	38.7 (12)	96.8 (30)	74.2 (23)	12.9 (4)	N/A	N/A
Lee (2010) [7]	13	13	63	46.2 (6)	100 (13)	92.3 (12)	0	N/A	N/A
Nonaka (2013) [8]	128	139	69.6	30.2 (42)	94.2 (131)	84.9 (118)	2.9 (4)	87.3	100
Ojima (2014) [9]	43	49	70	16.3 (8)	100 (49)	85.7 (42)	14.3 (7)	N/A	N/A
Yabuuchi (2019) * [10]	136	165	74	22.4 (37)	95.5 (150)	84.7 (133)	21.0 (33)	88.4	97.6
Present study	25	27	74	44.4 (12)	88.9 (24)	85.2 (23)	0	71.0	100

OS, overall survival, N/A, not applicable; * *En bloc* resection rate, R0 resection rate, and adverse events in Yabuuchi’s paper are not per lesion but per ESD procedure.

## Data Availability

Data are available on request due to restrictions, e.g., privacy or ethical.

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
