# Peer review of "Endoscopic Submucosal Dissection in Patients with Early Gastric Cancer in the Remnant Stomach"

_diagnostics, 2022, doi:10.3390/diagnostics12102480_

Round 1

Reviewer 1 Report

This is an interesting study to investigate the effective and safety of ESD to the EGCs in the remnant stomach. The conclusion is that ESD was effective and safe in the patients with EGCs in the remnant stomach. However, a lot of studies have published about the role of ESD in the patients with EGCs in the remnant stomach. This study didn’t have obvious new ideas. So, there are some questions what the author should further solve.

1.      Please discuss the highlights of this study compared with the previous study about the role of ESD in the patients with EGCs in the remnant stomach.

2.      How about the difference of ESD procedure such as operation time and complication in the patients with EGCs between the distal and proximal gastrectomy?

3.      Whether there are differences in the difficulties encountered in the operation of early gastric cancer in different parts such as the lesion in the greater curvature or the lesser curvature?

4.      For the patients with NCR, three patients were performed with piecemeal resection, what’s the reason?

Author Response

October 8, 2022

Dusan Vukelic,
Assistant Editor

Hajime Isomoto

Guest Editor

Diagnositcs

Dear Professor Vukelic, and Professor Isomoto:

Thank you for reviewing our manuscript titled "Endoscopic Submucosal Dissection in Patients with Early Gastric Cancer in the Remnant Stomach" by Mai Murakami, et al. for publication in Diagnositcs. This is an invited paper in the Special Issue "The Next Generation of Gastrointestinal Endoscopy".

We extend our appreciation to the reviewers for their insightful and constructive comments. We believe that these comments have helped us to improve the manuscript considerably. Our point-by-point responses to the reviewers are attached. The changes in the revised manuscript are shown in red font with underline by using the track changes mode in MS. We request that you review the revised manuscript, which we hope will now be acceptable for publication in Diagnositcs.

We look forward to hearing from you at your earliest convenience.

Sincerely,

Takuto Hikichi, MD, PhD.

Professor, Department of Endoscopy, Fukushima Medical University Hospital

e-mail: takuto@fmu.ac.jp

********************************************************************************

  • To Reviewer 1

This is an interesting study to investigate the effective and safety of ESD to the EGCs in the remnant stomach. The conclusion is that ESD was effective and safe in the patients with EGCs in the remnant stomach. However, a lot of studies have published about the role of ESD in the patients with EGCs in the remnant stomach. This study didn’t have obvious new ideas. So, there are some questions what the author should further solve.

(Respond)→We appreciate your significant comment.

We would like to respond to each of your review comments as follows.

Point 1. Please discuss the highlights of this study compared with the previous study about the role of ESD in the patients with EGCs in the remnant stomach.

(Response 1)→Thank you very much for your important suggestion.

In this study, the en bloc resection rate was 88.9%, which was lower than that reported previously, but the R0 resection rate was similar to that reported previously. Although the limitation of this study is the small number of cases, there were no cases of procedure-related adverse events, and the 5-year cause-specific survival rate was 100%, with no cases of gastric cancer death. We believe that these are the features of this study that are not found in previous reports. We have included these points in the Conclusion.

Point 2. How about the difference of ESD procedure such as operation time and complication in the patients with EGCs between the distal and proximal gastrectomy?

(Response 2)→Thank you for your very important comment.

The median diameter of the resected specimens was 34 mm (range 18-55 mm) and the median procedure time was 78 min (range: 42-210 min) in 21 RGC lesions after distal gastrectomy. On the other hand, RGC after proximal gastrectomy was 5 lesions in 3 patients, and in 2 of them, 2 lesions were resected en bloc as a single section. The median diameter of the resection specimens in the three patients with RGC after proximal gastrectomy was 44 mm (range 40-82 mm), and the median procedure time was 78 min (range: 42-210 min). Since only three patients underwent proximal gastrectomy, we did not consider it appropriate to perform a statistical analysis. However, a nonparametric test showed no difference in resection diameter (p=0.261), but a longer procedure time (p=0.011) for RGC after proximal gastrectomy. There were no procedure-related adverse events or complications for either gastrectomy. We have added this point to the Results section.

Point 3 Whether there are differences in the difficulties encountered in the operation of early gastric cancer in different parts such as the lesion in the greater curvature or the lesser curvature?

(Response 3)→Thank you for your valuable comment.

Although this study did not examine the difficulty of ESD technique by site, the retroflex scope manipulation was required for lesions located in the lesser curvature, and the residual stomach had a narrow working space, making it difficult to maneuver the scope in some cases. In addition, lesions located in the lesser curvature were affected by petz and fibrosis because they were sometimes in contact with the suture line. On the other hand, lesions located in the greater curvature had difficulty maintaining a good surgical field with blood when bleeding occurred intraoperatively.

These points have been added to the Results section.

Point 4. For the patients with NCR, three patients were performed with piecemeal resection, what’s the reason?

(Response 4)→Thank you very much for your important comment.

As described in the second paragraph of the discussion section, this was a case from 2006-2007, shortly after the introduction of ESD, not only because of patient factors but also because the surgeon's technique was not yet mature. The lesions were a cardia to fornix lesion, a greater curvature lesion in the upper gastric body, and a lesser curvature lesion in the antrum, respectively. The working space of each lesion was narrow, making dissection difficult, so segmental snaring resection was chosen.

Reviewer 2 Report

I thank the authors for the particular topic they have treated.

The paper is well structured. It shows that when, in time diagnosed and with a close follow-up, gastric cancer can be definitively cured.

Further more the endoscopic procedures, especially when performed by well-trained physician, has a crucial role especially in elderly or not well-fit patients.  

So I believe that the article is important from a scientific point of view and that it can be published.

I have only a little question:

in fig.2 (clinical course flow diagram) it is written: curative resection (n=20) and after follow-up (n=22). Is it an oversight?

Author Response

October 8, 2022

Dusan Vukelic,
Assistant Editor

Hajime Isomoto

Guest Editor

Diagnositcs

Dear Professor Vukelic, and Professor Isomoto:

Thank you for reviewing our manuscript titled "Endoscopic Submucosal Dissection in Patients with Early Gastric Cancer in the Remnant Stomach" by Mai Murakami, et al. for publication in Diagnositcs. This is an invited paper in the Special Issue "The Next Generation of Gastrointestinal Endoscopy".

We extend our appreciation to the reviewers for their insightful and constructive comments. We believe that these comments have helped us to improve the manuscript considerably. Our point-by-point responses to the reviewers are attached. The changes in the revised manuscript are shown in red font with underline by using the track changes mode in MS. We request that you review the revised manuscript, which we hope will now be acceptable for publication in Diagnositcs.

We look forward to hearing from you at your earliest convenience.

Sincerely,

Takuto Hikichi, MD, PhD.

Professor, Department of Endoscopy, Fukushima Medical University Hospital

e-mail: takuto@fmu.ac.jp

********************************************************************************

  • To Reviewer 2

I thank the authors for the particular topic they have treated. The paper is well structured. It shows that when, in time diagnosed and with a close follow-up, gastric cancer can be definitively cured. Furthermore, the endoscopic procedures, especially when performed by well-trained physician, has a crucial role especially in elderly or not well-fit patients. So I believe that the article is important from a scientific point of view and that it can be published.

(Respond)→We appreciate your kind comments.

We will respond to your review comments as follows.

Point 1. I have only a little question: in fig.2 (clinical course flow diagram) it is written: curative resection (n=20) and after follow-up (n=22). Is it an oversight?

(Response 1)→We apologize for the mistake. This was our mistake, as all curative resections are follow-ups, so there were 20 cases here. Thank you for pointing this out. The figure has been corrected.
